# Sputum Microbiome Composition in Patients with Squamous Cell Lung Carcinoma

**DOI:** 10.3390/life12091365

**Published:** 2022-09-01

**Authors:** Elizaveta Baranova, Vladimir Druzhinin, Ludmila Matskova, Pavel Demenkov, Valentin Volobaev, Varvara Minina, Alexey Larionov, Victor Titov

**Affiliations:** 1Department of Genetics and Fundamental Medicine, Kemerovo State University, Kemerovo 650000, Russia; 2Institute of Living Systems, Immanuel Kant Baltic Federal University, Kaliningrad 236041, Russia; 3Department of Microbiology, Tumor Biology and Cell Biology (MTC), 171 65 Stockholm, Sweden; 4Institute of Cytology and Genetics SB RAS, Novosibirsk 630090, Russia; 5Scientific Center for Genetics and Life Sciences, Sirius University of Science and Technology, Sochi 354340, Russia; 6Institute of Human Ecology, Federal Research Center of Coal and Coal Chemistry of Siberian Branch of the Russia Academy of Sciences, Kemerovo 650065, Russia; 7Kemerovo Regional Oncology Center, Kemerovo 654005, Russia

**Keywords:** lung cancer, squamous cell lung carcinoma, sputum microbiome, taxonomic composition, *Streptococcus*

## Abstract

Background: Recent findings indicate that the host microbiome can have a significant impact on the development of lung cancer by inducing an inflammatory response, causing dysbiosis, and generating genome damage. The aim of this study was to search for bacterial communities specifically associated with squamous cell carcinoma (LUSC). Methods: In this study, the taxonomic composition of the sputum microbiome of 40 men with untreated LUSC was compared with that of 40 healthy controls. Next-Generation sequencing of bacterial 16S rRNA genes was used to determine the taxonomic composition of the respiratory microbiome. Results: There were no differences in alpha diversity between the LUSC and control groups. Meanwhile, differences in the structure of bacterial communities (β diversity) among patients and controls differed significantly in sputum samples (pseudo-F = 1.53; *p* = 0.005). Genera of *Streptococcus*, *Bacillus*, *Gemella,* and *Haemophilus* were found to be significantly enriched in patients with LUSC compared to the control subjects, while 19 bacterial genera were significantly reduced, indicating a decrease in beta diversity in the microbiome of patients with LUSC. Conclusions: Among other candidates, *Streptococcus* (*Streptococcus agalactiae*) emerges as the most likely LUSC biomarker, but more research is needed to confirm this assumption.

## 1. Introduction

Interactions between the host and the commensal microbiota are complex and insufficiently understood. In cancer, diverse microbial ecosystems have been documented to induce metabolic changes in the tumor microenvironment, promote dysbiosis, directly induce oncogenic transformation, or modulate the immunotherapy response [1,2,3]. Comprehensive metagenomics approaches enable precise mapping of the tumor-associated microbiome and unveiling mechanisms of bacterial influence on cancer occurrence and progression [4]. In addition, recent efforts identified microbial signatures characteristic of certain cancer types, which may serve as tumor diagnostic biomarkers [5].

Lung cancer (LC) arises in the lung parenchyma or bronchi, and is annually diagnosed in approximately 1.2 million people worldwide with >1 million associated deaths during this period [6]. Although all forms of LC originate from epithelial cells of the airway mucosa, the current classification of LC includes several different histological types of this disease [7]. LC is usually divided into small cell lung cancer and non-small cell lung cancer (NSLC), which accounts for 85% of all bronchogenic tumors [8]. NSLC is further subdivided into large cell lung cancer, adenocarcinoma of the lung (AD), and lung squamous cell carcinoma (LUSC). LUSC accounts for about 30% of all NSLC cases. It is associated with a poor prognosis, and no targeted therapy is available so far [9].

The mortality rate from LUSC remains high, partly due to the lack of early detection of diagnostic biomarkers, including metagenomic ones. However, the search for bacteria associated with the risk of LC development has intensified tremendously in recent years, especially due to the wide application of the newest DNA sequencing technologies [10,11].

Previous studies have shown that changes in the number of specific microbiota taxa in bronchoalveolar lavage fluid, lung tissue, and saliva samples may be associated with LC, but results from these studies are largely inconsistent [12,13,14,15,16,17,18,19,20]. Another source of information on the composition of the respiratory tract microbiota is sputum, which has been poorly studied in patients with LC in general and particularly in those with LUSC [21,22,23,24,25]. Even though sputum is not reflecting the microbiome of any particular part of the respiratory tract, it can still be very useful as a metagenomic biomarker, since its collection is easy and non-invasive.

Different histological types of LC are characterized by different biological patterns, molecular markers, and treatment strategies [26]; however, very few studies have so far examined the relationship between the respiratory tract microbiome and individual histological types of LC.

In this report, we for the first time compare the taxonomic composition of the sputum microbiome in LUSC patients and healthy control donors, all residents of the Kuzbass region of Western Siberia.

## 2. Methods

### 2.1. Cohort Information

The composition of the sputum bacterial microbiome was studied in 40 patients with newly-diagnosed LUSC (male only, average age 59.9 ± 6.9 years) who were admitted to the Kemerovo Regional Oncology Center (Kemerovo, Russian Federation) and 40 healthy male donors, residents of Kemerovo (average age 54.0 ± 5.3 years). This material was collected from the period March 2018–August 2020. Active smokers were 75% and 55% of LUSC patients and control subjects, respectively. Smoking pack-years were not different between groups. For LUSC patients, the disease stage was determined in accordance with the TNM classification [27]: 18 patients (45%) were stage I–II, and 22 patients (55%) were stage III–IV. A questionnaire was filled out for each participant, containing information on place and date of the birth, living environment, occupation, exposure to occupational hazards, health status, dietary habits, and intake of medications (use of antibiotics at least four weeks prior to sampling), X-ray procedures, smoking and drinking status. For patients with LUSC, the results of clinical and histological analyses were additionally taken into account.

Inclusion criteria were adult males ≥40 years of age, willingness to participate in the study, donate sputum, and sign written informed consent. Exclusion criteria were any acute or chronic condition that would limit the ability of the patient to participate in the study, use of antibiotics within 4 weeks prior to collection, failure to obtain a sputum sample, or refusal to give informed consent.

All procedures undertaken were in accordance with the ethical standards of the Helsinki Declaration (1964 and amended in 2013) of the World Medical Association. All participants (patients and control subjects) were informed about the aim, methodology, and possible risks of the study; informed consent was signed by each donor. The design of this study was approved by the Ethics Committee of the Kemerovo State University (PROTOCOL CODE № 17/2021; 05.04.2021).

### 2.2. Sample Collection, Process, and Storage

To analyze the composition of the microbiome of the respiratory tract, sputum samples obtained from LC patients and control subjects were used. The sputum from patients was obtained prior to all diagnostic or therapeutic procedures. Sputum was collected on the first day of hospitalization. Before sputum collection, patients were asked to rinse their mouths. Sputum samples were collected non-invasively through participant-induced coughing (i.e., without induction) and represent the oropharyngeal secretion. Giemsa-stained cytological slide microscopy was used to test random sputum samples. The presence of columnar airway epithelial cells was confirmed. Samples were immediately placed in sterile plastic vials and frozen (−20 °C). Frozen samples were transported to the laboratory and stored at −80 °C.

### 2.3. DNA Extraction, 16S rRNA Gene Amplification and Sequencing

Procaryotic DNA was extracted using FastDNA Spin Kit For Soil (MP Biomedicals) based on the manufacturer’s recommendation. For each sample, 500 µL of sputum was used for DNA extraction. The DNA concentration was monitored using Qubit^®^ dsDNA Assay Kit in Qubit^®^ Flurometer (Life Technologies, Carlsbad, CA, USA). Eighty 16S rRNA gene amplicon libraries were prepared by PCR amplification of an approximate 467 bp region within the hypervariable (V3–V4) region of the 16S rRNA gene in bacteria, from 50 ng of each of the extracted and purified DNA from sputum samples, respectively, according to the Illumina 16S metagenomic sequencing library protocol. PCR was initially performed with broad-spectrum 16S rRNA primers (forward primer: 5′-TCGTCGGCAGCGTCAGATGTGTATAAGAGACAGCCTACGGGNGGCWGCAG-3′, and reverse primer: 5′-GTCTCGTGGGCTCGGAGATGTGTATAAGAGACAGGACTACHVGGGTATCTAATCC-3′), using BioMaster Hi-Fi LR 2X ReadyMix DNA polymerase (BiolabMix company, Novosibirsk, Russia). Cycle conditions were 94 °C (3 min 30 s), then 25 cycles of 94 °C (30 s), 55 °C (30 s), 68 °C (40 s), then a final extension of 68 °C (5 min). Libraries were purified using Agencourt AMPure XP beads (Beckman Coulter, Bray, Houston, TX, USA) according to the Illumina 16 S metagenomic sequencing library protocol. Dual indices and Illumina sequencing adapters from the Illumina Nextera XT index kits v2 B and C (Illumina, San Diego, CA, USA) were added to the target amplicons in a second PCR step using BioMaster Hi-Fi LR 2× ReadyMix DNA polymerase (BiolabMix company, Novosibirsk, Russia). The primer sequence was taken from the recommended library preparation protocol for sequencing on the MiSeq platform https://support.illumina.com/documents/documentation/chemistry_documentation/16s/16s-metagenomic-library-prep-guide-15044223-b.pdf (accessed on 27 November 2013). Cycle conditions were 94 °C (3 min 30 s), then 8 cycles of 94 °C (30 s), 55 °C (30 s), 68 °C (40 s), then a final extension of 68 °C (5 min). Libraries were again purified using Agencourt AMPure XP beads (Beckman Coulter, Bray, Houston, TX, USA) according to the Illumina 16 S metagenomic sequencing library protocol. Sample PCR products were then pooled in equimolar amounts, purified using AMPure XP Beads (Beckman Coulter, Bray, Houston, TX, USA), and then quantified using a fluorometer (Quantus Fluorometer dsDNA (Promega, Madison, WI, USA). Molarity was then brought to 4 nM, the libraries were denatured, and then diluted to a final concentration of 8 pM with a 10% PhiX spike buffer for sequencing on the Illumina MiSeq platform [28].

### 2.4. Taxonomy Quantification Using 16S rRNA Gene Sequences and Statistical Methods

The processing of the resulting sequence data was conducted using the QIIME2 software [29]. A quality check was carried out and a sequence library was generated. The sequences were combined into operational taxonomic units (OTUs) based on a 99% nucleotide similarity threshold using the Greengenes reference sequences library (versions 13–8) and SILVA (version 132), followed by the removal of singletons (OTUs containing only one sequence). The total diversity of prokaryotic communities (alpha diversity) of sputum was estimated by the number of allocated OTU (analogue of species richness) and Shannon indices (H = Σpi ln pi, pi—part of i-sh species in a community). When calculating sample diversity indices, 1045 sequences were normalized (the minimum number of received sequences per sample). The variation in the structure of the bacterial community of different samples (beta diversity) was analyzed using Bray-Curtis dissimilarity metrics [30]—a method common in microbial ecology that estimates the difference between communities based on the abundance relationships of the taxa present in the samples.

In addition, to assess the significance of differences in the relative percentage of individual bacterial taxa in sputum, the Mann-Whitney U test was used. Spearman’s correlation coefficient was used to calculate correlations. Calculations were performed using the software package STATISTICA.10, Statsoft, Tulsa, OK, USA. The False Discovery Rate (FDR) correction was used to assess the significance of differences in the relative percentages of individual bacterial taxa taking into account multiple comparisons. Multiple linear regression (MLR) was performed to predict the relationship between the relative abundance of individual bacteria in LUSC patients’ sputum and lifestyle/disease factors.

## 3. Results

Here we profiled the composition of the sputum bacterial microbiome across 40 patients with LUSC and 40 healthy donors, all residents of Kemerovo. We have used a large-scale approach to sequence the 16S rRNA V3–V4 region of the bacterial genomes purified from the sputum samples from the compared groups in the study. A summary of the demographic information regarding LUSC and control subjects is shown in Table 1. There were differences in mean age between patients and control (*p* < 0.05). Both LUSC and healthy control groups were sex-matched, and had no differences in living environment, alcohol consumption, and smoking pack years.

For the LUSC group, the average number of analyzed sequences was 76,776 (range: 9694−181,146). For the healthy control group, the average number of analyzed sequences was 72,613 (range: 12,537−160,232). We identified a total of 11 bacterial phyla with relative frequencies above 0.1%. The prevailing phyla in our dataset were *Firmicutes*, *Bacteroidetes*, *Actinobacteria,* and *Proteobacteria* (Figure 1), consistent with results from previous studies [21,22,31,32].

Regarding alpha diversity, neither the number of allocated OTUs nor the Shannon indices showed significant differences between LUSC and control groups. Overall, the bacterial communities were fairly diverse in the two groups as indicated by the Shannon index at the genus level (5.267 in LUSC vs. 5.439 in control groups). This suggests that any changes in the sputum microbiome of the LUSC are not large-scale shifts in the bacterial community.

Differences in the structure of bacterial communities in sputum samples of lung cancer patients and healthy subjects are shown in Figure 2. The first two principle components explained 14.47 and 7.182% of the total variation. Compositional similarity within the phylum-level taxa was displayed among individual samples using the bar plot (Figure 1). The PERMANOVA (Adonis) test using the difference matrix, constructed by the Bray-Curtis method, showed a significant difference in the prokaryotic communities in sputum from healthy subjects and patients with LUSC (pseudo-F = 1.53; *p* = 0.005).

We then compared frequencies of major bacteria phyla in our sputum specimens. Samples from LUSC patients revealed a significant increase in the representatives of *Firmicutes* phylum as compared to control subjects (56.77 ± 15.29 vs. 47.34 ± 10.65 %, respectively; *p* = 0.004); in contrast, the other four major bacterial phyla (*Bacteroidetes*, *Fusobacteria*, *TM7*, and *Spirochaetes*), were overrepresented in the sputum of healthy subjects in comparison with that from LUSC patients (Figure 3).

Analysis of the composition of microbial communities in LUSC and control sputum enabled us to annotate the core microbiome of our sputum samples, which consisted of 67 genera and 32 species. Bacterial genera and species significantly different between groups are listed in Table 2 and Table 3 (23 and 17, respectively, after FDR correction). We observed a considerable variation in the percentages of all genera and species presented.

The sputum of LUSC patients was characterized by a significant enrichment of the following genera (ranked by percentage): *Streptococcus* (36.26 ± 20.02 vs. 18.93 ± 10.43; *p* = 0.00001); *Bacillus* (3.55 ± 2.9 vs. 1.84 ± 1.93; *p* = 0.003); *Gemella* (3.6 ± 2.89 vs. 2.01 ± 2.01; *p* = 0.004) and *Haemophilus* (1.27 ± 8.07 vs. 0.11 ± 0.36; *p* = 0.003). At the same time, members of the 19 genera in Table 2 were significantly overrepresented in the microbiome of the control group in comparison to the patients with LUSC.

At the species level, only the *Streptococcus agalactiae* was significantly higher in the sputum of patients compared to control subjects (35.47 ± 20.19 vs. 19.11 ± 10.06; *p* = 0.00004). Representatives of the other 17 species were significantly more common in the microbiome of healthy controls in comparison to the patients with LUSC, as shown in Table 3.

We found no specific association of any bacterial taxon in the sputum with the age of patients or control donors participating in the study.

The influence of smoking status on the microbiota composition in patients with LUSC and control subjects was also investigated. For LUSC patients, no significant difference was found in the bacterial genera or species in sputum between smokers and nonsmokers. Controls differing in smoking status revealed a significant difference in the occurrence of several genera and species in the sputum. Control group smokers (Figure 4) had less *Neisseria* than non-smokers (0.56 ± 1.16% vs. 3.94 ± 5.63%; *p* = 0.00006); *Fusobacterium* (1.4 ± 1.55% vs. 3.39 ± 3.01%; *p* = 0.02); *Prevotella nigrescens* (0.35 ± 1.38% vs. 0.52 ± 0.68%; *p* = 0.01) and *Peptostreptococcus Anaerobius* (0.04 ± 0.1% vs. 0.39 ± 0.71%; *p* = 0.02). At the same time, control group smokers had more *Streptobacillus* in their sputum compared to nonsmokers (3.62 ± 2.8% vs. 1.92 ± 2.28%; *p* = 0.03).

Comparison of the total composition of the microbiome in patients with different stages of LUSC (I–II and III–IV), as well as between subgroups with different localization of the primary tumor, revealed no differences.

Conditional logistic regression models adjusted for age, smoking status, alcohol consumption status, living environment, occupational exposure, family cancer history, chronic diseases (heart and vessels, bronchitis, COPD, stomach, diabetes and obesity) and the phyla (*Streptococcus*, *Bacillus*, *Gemella* and *Haemophilus*) were constructed. In these models, heart and vessels diseases (*p* = 0.0001), bronchitis (*p* = 0.008), COPD (*p* = 0.003), and presence of *Streptococcus* (*p* = 0.009) were strongly associated with LUSC as compared to healthy subjects.

## 4. Discussion

The respiratory tract microbiome is closely linked to the onset of lung diseases, including LC. It has been previously shown that there are changes in the microecology of the lungs in patients with lung cancer compared to healthy subjects. In addition, the abundance of certain bacterial species correlates with pathology, suggesting their potential use as microbial markers for the detection of lung cancer. However, until now, the composition of the lung microbiome in patients with different histological types of lung cancer has not been determined.

In this study, we examined the difference between the microbiome of sputum samples from patients with LUSC and healthy controls. In general, the sputum microbiota in men with lung cancer had a significant decrease in beta diversity, which is consistent with the results of previous studies [13,21,33,34]. At the level of bacterial phyla, the most notable finding in our patients with LUSC was an abundance of *Firmicutes* to the detriment of *Proteobacteria*. The dominance of *Proteobacteria* in healthy lung microbiota was also detected by others [18,35]. In a pairwise comparison, representatives of four bacterial phyla (*Bacteroidetes*, *Fusobacteria*, *TM7,* and *Spirochaetes*) and the 19 genera shown in Table 2 were significantly enriched in healthy control samples as compared to LUSC patients. On the other hand, we found that *Streptococcus,* belonging to the *Firmicutes* phylum, demonstrated the highest abundance in LUSC patients in comparison with controls. Two other genera (*Bacillus* and *Gemella*) from the *Firmicutes* type, and a representative of *Proteobacteria*—*Haemophilus*, were also overrepresented in the sputum of LUSC patients compared to controls. We believe that all four genera may be considered potential bacterial biomarkers of LUSC.

An increased prevalence of *Streptococcus* in the sputum of patients with lung cancer has previously been reported in several publications [21,22,26]; however, a high abundance of the *Bacillus*, *Gemella,* and *Haemophilus* genera were not previously reported. Indeed, a recent study using ddPCR found a significant increase in *Streptococcus* load in the sputum of seven patients with LUSC compared with ten control patients [36]. Interestingly, at the same time, a significant increase in *Veillonella* was found in the sputum of the same patients in comparison to control participants. In our study, however, representatives of this bacterial genus, were more evidently enriched in controls than in LUSC patients (Table 2). Finally, the amount of *Haemophilus* in the sputum of patients and controls was almost equal [31], while in our cohort of patients this facultative anaerobe was significantly more common in LUSC patients as compared to healthy donors. Another recent study of the respiratory microbiome (saliva and bronchial biopsy specimens) in 25 patients with central lung cancer from Spain [37] found a significant increase in *Streptococcus*, *Rothia*, *Gemella,* and *Lactobacillus*, which partially agrees with our results (for *Streptococcus* and *Gemella*). Thus, it appears that *Streptococcus* is a major bacterial marker in the airways associated with lung cancer, although it could depend on different histopathological types and stages of this disease. For example, it was reported that *Streptococcus* and *Neisseria* were predominant in the sputum of patients with lung adenocarcinoma, while *Streptococcus* and then *Veillonella* dominated the microbiome of LUSC patients, while *Neisseria* and, to a lesser extent, *Streptococcus*, were the most frequently found genera in the sputum of small cell lung cancer patients [21].

A comparison of sputum microbiome composition in subgroups of LUSC patients with different TNM stages, central or peripheral tumor localization, and smoking status, revealed no significant differences in bacterial content. However, in the group of healthy donors, we observed a prominent decrease in *Neisseria* in the sputum of smokers compared to non-smokers (Figure 4), which is consistent with previously published results [38]. The effect of smoking on the sputum microbiota remains unclear, according to the latest published data [39], and requires further study.

As shown in Table 3, *Streptococcus agalactiae* was the only bacterial species that significantly increased in patient sputum, according to sequencing data and analysis of two databases (Greengenes and SILVA). It should be noted that Streptococcal species are difficult to identify using 16S rRNA gene sequencing alone, and requires further validation using ddPCR. Previous studies have reported the prevalence of *Streptococcus viridans* in the sputum of patients with lung cancer [17]. In our study, *Streptococcus agalactiae* was the most frequently found bacteria in the sputum of both LUSC patients and controls, and its significant increase in LC patients suggests its utility as a possible biomarker, similar to *Streptococcus gallolyticus subsp*. in colorectal carcinoma [40]. *Streptococcus agalactiae* (also known as GBS) is an important opportunistic species that can cause pneumonia, sepsis, and meningitis in newborns and in immunocompromised subjects [41,42]. Cases of invasive GBS infections are frequently reported in the elderly and immunocompromised adults, including those with diabetes mellitus, alcoholism, and cancer [43]. In the respiratory tract, GBS occasionally contributes to community-acquired pneumonia and empyema in adults [44]. When GBS causes a pulmonary infection, it is usually defined as part of polymicrobial pneumonia [45]. GBS bacteria effectively attach to pulmonary epithelial cells and are capable of invasion. This is initiated by their attachment to extracellular matrix components such as agglutinin, fibronectin, fibrinogen, and laminin, which facilitates their attachment to host cell surface proteins, such as integrins. Thus, the invasive potential of GBS is influenced by changes in the surface proteome of host cells, which can be caused by various lung pathologies [46]. The molecular mechanisms of cytopathology caused by GBS bacteria in patients are currently being intensively studied. It was shown that GBS induces the generation of reactive oxygen species and loss of mitochondrial membrane potential [47]. In human endothelial cells, reactive oxygen species are generated via the NADPH oxidase pathway, which is accompanied by cytoskeletal reorganization through the PI3K/Akt pathway, and is generally associated with pathogen penetration, providing evidence for the involvement of oxidative stress in the pathogenesis associated with *S. agalactiae* [48].

Several limitations of this study should be noted. First, our study with 80 samples may not be powerful enough. Our results require confirmation in independent large-scale studies to further understand the role of the sputum microbiota in the development of lung cancer. Second, only men were included in the present study, so women with LUSC should be studied further. Finally, at this stage of the study, we cannot unequivocally identify the specific Streptococcal species whose presence in patients’ sputum is elevated compared with controls. Further analysis using ddPCR will eliminate this limitation.

## 5. Conclusions

In this report, we used mass parallel sequencing of bacterial 16S ribosomal genes to compare the taxonomic composition of the sputum microbiome of patients with LUSC and healthy donors. It was found that the bacterial taxonomic groups detected in the microbiome of patients were significantly different compared to controls. The sputum of patients with LUSC contains significantly more members of the genera *Streptococcus*, *Bacillus*, *Gemella,* and *Haemophilus*. *Streptococcus* (*Streptococcus agalactiae*) is the most likely LUSC biomarker from this list, but more research is still required to validate this assumption.

In order to consider these bacteria as biomarkers for the risk of LUSC development, it is necessary to have information about their population dynamics in the respiratory microbiome from health to lung malignancy. This can be solved, for example, by forming a database of the respiratory microbiome in healthy individuals over a long period of time. Another possible and more accessible approach is to study the composition of the microbiome in the sputum of patients with chronic inflammatory diseases of the lungs. A recent study showed increased numbers of *Streptococci* in airway microbiome samples from patients with idiopathic pulmonary fibrosis and COPD. It is important, in this regard, that our logistic regression models showed a significant relationship between an increase in abundance of *Streptococcus* and chronic inflammatory lung diseases, such as bronchitis and COPD in patients with LUSC. Thus, future studies to establish the role of bacteria as biomarkers of LC should examine the composition of the sputum microbiome in these and other non-malignant lung diseases.

## Figures and Tables

**Figure 1 life-12-01365-f001:**
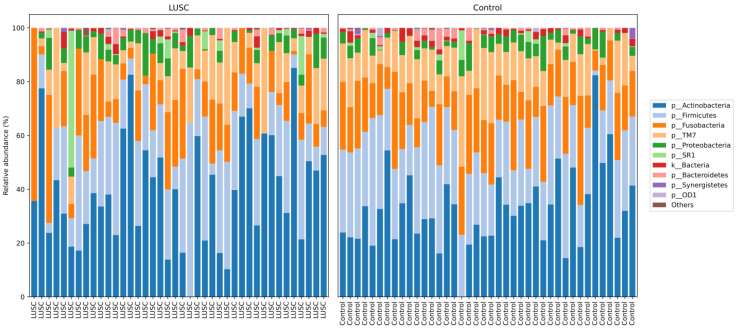
Taxonomic structure of the sputum microbiomes from LUSC patients (*n* = 40) and control subjects (*n* = 40) at the phyla level.

**Figure 2 life-12-01365-f002:**
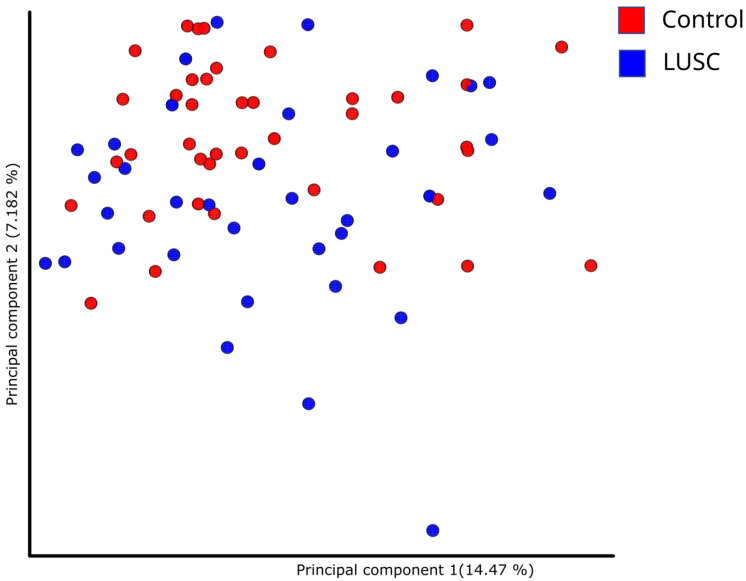
Principal component analysis demonstrating phylogenetic similarity of prokaryotic sputum communities in LUSC patients and control subjects.

**Figure 3 life-12-01365-f003:**
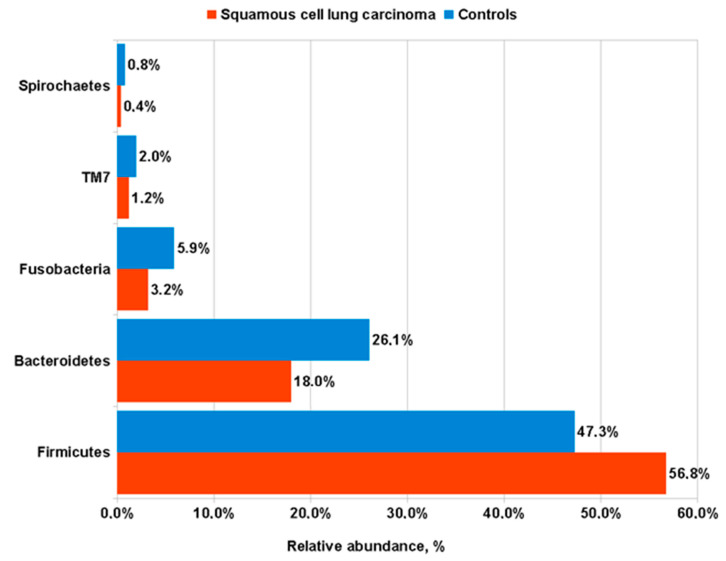
Frequencies of major bacterial phyla in the sputum of LUSC patients compared to control subjects.

**Figure 4 life-12-01365-f004:**
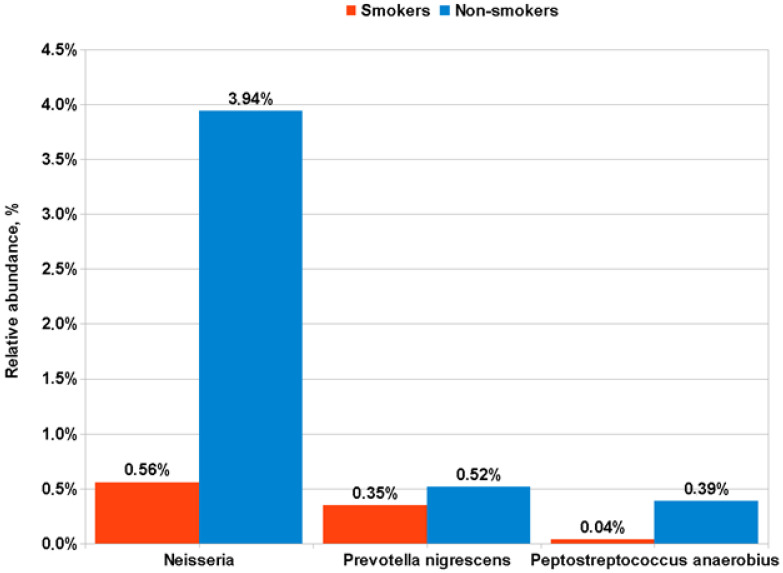
Differences in the representation of bacterial taxa in the sputum of smokers and non-smoker control subjects.

**Table 1 life-12-01365-t001:** Demographic and clinical data of patients with squamous cell carcinoma and healthy controls.

Variables	Squamous Cell Lung Carcinoma, *n* = 40	Healthy Kemerovo Residents (Control), *n* = 40
Age (years) (mean ± SD)	59.9 ± 6.9 *	54.0 ± 5.3
Living environment (%):		
City	67.0	85.0
Village	33.0	15.0
Occupational exposure (%):		
Yes	45.0	0
No	65.0	100
Smoking status (%):		
Non-smokers	25.0	45.0
Smokers	75.0	55.0
Smoking pack-years (mean ± SD)	37.13 ± 12.7	32.91 ± 12.62
Alcohol status (%):		
Non-drinker	25.0	10.0
Rare drinker (1–2 times per month)	52.0	45.0
Medium drinker (1–2 times a week)	23.0	45.0
Diet (%):		
Vegetarian	0	0
Non-vegetarian	100	100
Family cancer history (%):		
Yes	28.0	40.0
No	72.2	60.0
Chronic diseases (%):		
Heart and vessels	90.0	22.5
Bronchitis	32.5	7.5
COPD	45.0	0
Stomach (gastritis, ulcer)	17.5	15.0
Diabetes	0	2.5
Obesity	20.0	0
TNM ^#^ (%):		-
I, II	45.0	
III, IV	55.0	
Distant metastasis (%):		-
Yes	5.0	
No	95.0	
Tumor localization (%):		-
Central lung cancer	30	
Peripheral lung cancer	42.5	
Mixed lung cancer	7.5	
Bronchial cancer	20	

* *p* < 0.05: significant differences in comparison with controls; ^#^ tumor-node-metastasis.

**Table 2 life-12-01365-t002:** Differences in bacterial genera in the core microbiome of sputum from LUSC patients and healthy subjects. Mann-Whitney U test.

Genus	Squamous Cell Lung Cancer, % (*n* = 40)	Controls, % (*n* = 40)	*p* *
*Streptococcus*	36.26 ± 20.02	18.93 ± 10.43	0.00001
*Prevotella (f.Prevotellfceae)*	10.86 ± 7.03	17.89 ± 7.37	0.00003
*Veillonella*	6.6 ± 7.45	11.17 ± 6.22	0.00009
*Anaerosinus*	6.57 ± 7.69	10.91 ± 6.33	0.0003
*Gemella*	3.6 ± 2.89	2.01 ± 2.01	0.004
*Bacillus*	3.55 ± 2.96	1.84 ± 1.93	0.003
*Haemophilus*	2.2 ± 10.32	0.13 ± 0.4	0.003
*Selenomonas*	1.68 ± 2.82	4.36 ± 3.49	0.00003
*Megasphaera*	1.38 ± 2.91	2.44 ± 2.27	0.005
*Streptobacillus*	1.14 ± 1.46	2.85 ± 2.69	0.002
*Atopobium*	1.22 ± 1.79	1.66 ± 1.47	0.02
*Leptotrichia*	1.13 ± 1.35	2.63 ± 2.49	0.008
*Treponema*	0.43 ± 0.76	0.73 ± 1.01	0.002
*Lachnoanaerobaculum*	0.39 ± 0.53	0.65 ± 0.65	0.03
*Porphiromonas*	0.35 ± 0.75	0.93 ± 1.3	0.002
*Parvimonas*	0.42 ± 0.83	0.83 ± 1.06	0.002
*Stomatobaculum*	0.39 ± 0.42	0.9 ± 0.81	0.003
*Vestibaculum*	0.35 ± 1.1	0.86 ± 1.45	0.005
*Catonella*	0.09 ± 0.32	0.14 ± 0.25	0.03
*Filifactor*	0.06 ± 0.16	0.21 ± 0.33	0.003
*Mycoplasma*	0.04 ± 0.09	0.28 ± 0.72	0.02
*Moriella*	0.05 ± 0.18	0.36 ± 0.83	0.02
*Cardiobacterium*	0.02 ± 0.1	0.04 ± 0.1	0.03

Note: * *p* Value lesser than FDR corrected *p*.

**Table 3 life-12-01365-t003:** Distribution of bacterial species in the core microbiome of sputum from LUSC patients and healthy subjects. Mann-Whitney U test.

Species	Squamous Cell Lung Cancer, *n* = 40	Controls, *n* = 40	*p* *
*Streptococcus agalactiae*	35.47 ± 20.19	19.11 ± 10.06	0.00004
*Anaerosinus glycerini*	4.8 ± 6.31	10.19 ± 6.81	0.0003
*Selenomonas bovis*	1.44 ± 2.62	4.36 ± 4.46	0.00001
*Prevotella histicola F0411*	1.42 ± 2.27	2.84 ± 3.2	0.02
*Atopobium rimae*	1.29 ± 1.77	1.68 ± 1.46	0.03
*Megasphaera Micronuciformis*	1.23 ± 2.88	2.39 ± 2.34	0.001
*Lachnoanaerobaculum orale*	0.36 ± 0.5	0.69 ± 0.64	0.01
*Vestibaculum illigatum*	0.29 ± 0.98	0.91 ± 1.46	0.003
*Rothia dentocariosa ATCC 17931*	0.23 ± 0.6	0.53 ± 0.8	0.02
*Prevotella sp.oral clone DO014*	0.16 ± 0.44	0.52 ± 0.6	0.0007
*Porphyromonas endodontalis*	0.14 ± 0.34	0.93 ± 1.3	0.003
*Prevotella intermedia*	0.14 ± 0.34	0.47 ± 0.96	0.04
*Moryella indoligenes*	0.06 ± 0.19	0.4 ± 0.895	0.03
*Prevotella nigrescens*	0.09 ± 0.22	0.43 ± 1.11	0.02
*Oribacterium Sinus*	0.07 ± 0.22	0.26 ± 0.51	0.03
*Leptotrichia sp. oral clone EI013*	0.06 ± 0.18	0.24 ± 0.54	0.0008
*Filifactor alocis ATCC 35896*	0.06 ± 0.16	0.21 ± 0.33	0.007

Note: * *p* Value lesser than FDR corrected *p*.

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
