# Peer review of "Sputum Microbiome Composition in Patients with Squamous Cell Lung Carcinoma"

_life, 2022, doi:10.3390/life12091365_

Round 1
Reviewer 1 Report
This study compared the taxonomic composition of the sputum microbiome of patients with squamous cell carcinoma (LUSC) and healthy donors. The research design is innovative and clinically useful. Besides, there are some major and minor questions need to be solved. Please revise according to the following questions.
Major points are:
1.In Table 1, the lung inflammatory diseases such as bronchitis and COPD in the LUSC group were significantly more than those in the control group. Considering the close relationship between lung inflammation and lung microorganisms, subgroup analysis or enlarged samples for matching is recommended.
2.How to eliminate the interference of microbiome in upper respiratory tract samples? Are lower respiratory tract specimens such as BALF more appropriate?
Minor points are:
1.The font size in the article is not uniform.
2.Normal healthy people rarely cough up sputum. Please describe the method of sputum collection concretely. Please specify whether hypertonic saline is used to induce sputum.
3.In Figure 2, Principal component 1 (14.47%) and Principal component 2 (7.182%) were not fully explained in the results.
4.It is suggested to add the limitation analysis in the Discussion.
Author Response
The authors thank the Reviewer for his attention to our manuscript.
English language and style are fine/minor spell check required
Reviewer 1 asked to correct the English. We have done it.
LM: English has been corrected
We have also added this sentence to the Acknowledgments section: We thank Assoc. Prof. Dr. Gösta Winberg, MD, PhD, a graduate of the Department of Linguistics at Stockholm University, for his help in correcting English.
Comments
Major points are:
- In Table 1, the lung inflammatory diseases such as bronchitis and COPD in the LUSC group were significantly more than those in the control group. Considering the close relationship between lung inflammation and lung microorganisms, subgroup analysis or enlarged samples for matching is recommended.
As recommended by the reviewer, we performed additional analysis. The representation of types, genera and species of bacteria in sputum was compared in three subgroups of LUSC patients: 1 - do not have concomitant inflammatory lung diseases; 2 - have a history of chronic bronchitis; 3 - have a history of COPD. We found no significant differences in bacterial content between these subgroups. For example, between subgroups 1 and 2 there were differences in the representation of Capnocytophaga (P= 0.02) and Prevotella pallens (P= 0.03). Between subgroups 1 and 3 there were differences in the representation of Prevotella nigrescens (P= 0.03). However, the use of FDD correction for multiple comparisons showed that these differences are not significant. In future studies, we plan to separately study the composition of the sputum microbiome in patients with chronic bronchitis, as well as in patients with COPD.
- How to eliminate the interference of microbiome in upper respiratory tract samples? Are lower respiratory tract specimens such as BALF more appropriate?
Bronchoalveolar lavage does reflect the composition of the lower respiratory tract microbiota (although contamination of these samples cannot be completely ruled out), but sputum samples are important as a noninvasive source of material, even though they may be contaminated with upper respiratory bacteria. Ultimately, it does not matter which specimens (sputum or BALF) are used as long as they contain bacteria that reliably distinguish patients from healthy individuals.
Minor points are:
- The font size in the article is not uniform.
The font size in the article is now made uniform.
- Normal healthy people rarely cough up sputum. Please describe the method of sputum collection concretely. Please specify whether hypertonic saline is used to induce sputum.
Sputum samples were collected non-invasively through participant-induced coughing (i.e., without induction). We have not used hypertonic saline for cough induction. Some volunteers were unable to cough up sputum and in these cases they were excluded from the study. We indicated this in the exclusion criteria in the Cohort information section.
- In Figure 2, Principal component 1 (14.47%) and Principal component 2 (7.182%) were not fully explained in the results.
We've added an explanation to the Results section: « Differences in the structure of bacterial communities in sputum samples of lung cancer patients and healthy subjects are shown in Fig. 2. The first two principle components explained 14.47 and 7.182% of the total variation. Compositional similarity within the phylum-level taxa was displayed among individual samples using the bar plot (Fig. 1)»
- It is suggested to add the limitation analysis in the Discussion.
We have added reasoning about the limitations of our approach to the Discussion section. ” Several limitations of this study should be noted. First, our study with 80 samples may not be powerful enough. Our results require confirmation in independent large-scale studies to further understand the role of the sputum microbiota in the development of lung cancer. Second, only men were included in the present study, so women with LUSC should be studied further. Finally, at this stage of the study, we cannot unequivocally identify the specific Streptococcal species whose presence in patients' sputum is elevated compared with controls. Further analysis using ddPCR will eliminate this limitation.”
Reviewer 2 Report
Sputum microbiome composition in patients with squamous 1 cell lung carcinoma by Baranova et al is interesting. However, it needs some clarifications.
Minor comments.
1. Figure 3. The authors show that Frequencies of major bacterial phyla in the sputum of LUSC patients, what is the sequence homology of these microbes when compared to the known data-base?
2. Table 2. what is the sequence homology of these microbes when compared to known data-base?
Author Response
The authors thank the Reviewer for his attention to our manuscript.
Minor comments.
- Figure 3. The authors show that Frequencies of major bacterial phyla in the sputum of LUSC patients, what is the sequence homology of these microbes when compared to the known data-base?
The sequence homology of all defined operational taxonomic units is consistent
with the Greengenes reference sequences library (versions 13–8) and SILVA
(version 132) databases.
- Table 2. what is the sequence homology of these microbes when compared to known data-base?
The sequence homology of all defined operational taxonomic units is consisten with the Greengenes reference sequences library (versions 13–8) and SILVA (version 132) databases.
Round 2
Reviewer 1 Report
All suggestions offered last time are accepted in the revised manuscript. Essential information has been added to results and discussion.